# Discovery of Epipodophyllotoxin-Derived B_2_ as Promising *Xoo*FtsZ Inhibitor for Controlling Bacterial Cell Division: Structure-Based Virtual Screening, Synthesis, and SAR Study

**DOI:** 10.3390/ijms23169119

**Published:** 2022-08-14

**Authors:** Ying-Lian Song, Shuai-Shuai Liu, Jie Yang, Jiao Xie, Xiang Zhou, Zhi-Bing Wu, Li-Wei Liu, Pei-Yi Wang, Song Yang

**Affiliations:** State Key Laboratory Breeding Base of Green Pesticide and Agricultural Bioengineering, Key Laboratory of Green Pesticide and Agricultural Bioengineering, Ministry of Education, Center for R & D of Fine Chemicals of Guizhou University, Guiyang 550025, China

**Keywords:** natural products (NPs), structure-based virtual screening (SBVS), *Xanthomonas oryzae* pv. *oryzae* (*Xoo*), FtsZ inhibitors, plant bacterial diseases

## Abstract

The emergence of phytopathogenic bacteria resistant to antibacterial agents has rendered previously manageable plant diseases intractable, highlighting the need for safe and environmentally responsible agrochemicals. Inhibition of bacterial cell division by targeting bacterial cell division protein FtsZ has been proposed as a promising strategy for developing novel antibacterial agents. We previously identified 4′-demethylepipodophyllotoxin (DMEP), a naturally occurring substance isolated from the barberry species *Dysosma versipellis*, as a novel chemical scaffold for the development of inhibitors of FtsZ from the rice blight pathogen *Xanthomonas oryzae* pv. *oryzae* (*Xoo*). Therefore, constructing structure−activity relationship (SAR) studies of DMEP is indispensable for new agrochemical discovery. In this study, we performed a structure−activity relationship (SAR) study of DMEP derivatives as potential *Xoo*FtsZ inhibitors through introducing the structure-based virtual screening (SBVS) approach and various biochemical methods. Notably, prepared compound **B_2_**, a 4′-acyloxy DMEP analog, had a 50% inhibitory concentration of 159.4 µM for inhibition of recombinant *Xoo*FtsZ GTPase, which was lower than that of the parent DMEP (278.0 µM). Compound **B_2_** potently inhibited *Xoo* growth in vitro (minimum inhibitory concentration 153 mg L^−1^) and had 54.9% and 48.4% curative and protective control efficiencies against rice blight in vivo. Moreover, compound **B_2_** also showed low toxicity for non-target organisms, including rice plant and mammalian cell. Given these interesting results, we provide a novel strategy to discover and optimize promising bactericidal compounds for the management of plant bacterial diseases.

## 1. Introduction

Plant diseases caused by phytopathogenic bacterium represent crucial threats to plant health and the productivity of agriculture crops [1,2,3]. The human population is predicted to increase to 10 billion by the year 2100, which will require a doubling or tripling of current agricultural productivity to ensure that adequate food supplies are maintained [4,5,6]. Modern agriculture has benefited substantially from the use of agrochemicals, but many, especially traditional agrochemicals, are hazardous to both the environment and human health [7,8]. For instance, bismerthiazol (BT), a commercial bactericide active against *Xanthomonas oryzae* pv. *oryzae* (*Xoo*), exhibits subchronic and chronic toxicity in humans upon oral consumption [9]. These drawbacks have highlighted the urgent need for safer and more environmentally responsible pesticides.

Natural product-based pesticides have many potential advantages over synthetic compounds, including lower toxicity and easier and more environmentally friendly degradation. Consequently, there has been exponential growth in the development of new agrochemicals originating from natural products and their derivatives, which are seen as an effective panacea for integrated pest management [10,11,12,13]. For instance, some natural *β*-methoxyacrylic acid fungicides and their synthetic strobilurin derivatives are extensively used to control fungal pathogens [14]. Similarly, natural pyrethrum and synthetic pyrethroids are used commercially to control insects [15]. These examples illustrate the potential for natural products and their derivatives to be developed as new pesticides.

4′-Demethylepipodophyllotoxin (DMEP) is an aryltetralin cyclolignan isolated from the barberry species *Dysosma versipellis* and represents a structural framework for many compounds shown to display various bioactivities [16,17,18]. For example, structural modification of DMEP has yielded many anticancer agents, including etoposide and teniposide [16,19], and various DMEP analogs and derivatives with insecticidal activity have been developed and shown to successfully control insect pests in recent years [20,21,22]. We recently employed the framework of DMEP to develop inhibitors of bacterial FtsZ, a tubulin homolog that possesses GTPase activity, that have bactericidal activity and control bacterial leaf blight of rice [23]. The results of that study suggested a new drug discovery strategy and application for DMEP to develop potent FtsZ-targeting compounds for controlling intractable bacterial diseases of plants.

In the present study, we utilized a structure-based virtual screening strategy to guide the design of candidate DMEP-derived compounds with bactericidal properties [24]. Structure-based virtual screening is an increasingly common and prominent strategy in drug discovery and facilitates the design of synthesizable and novel chemical structures with certain molecular targets and bioactivities. For example, candidate agents targeting the kinase discoidin domain receptor 1, which has been implicated in many human diseases, were identified using the DrugSpaceX platform (https://drugspacex.simm.ac.cn/, accessed on 1 October 2021), which catalogs features such as drug-likeness, synthesizability, diversity, and novelty of compounds within a three-dimensional chemical space [25]. In the present study, we designed and synthesized a panel of DMEP derivatives and evaluated their ability to inhibit recombinant *Xoo*FtsZ GTPase activity and *Xoo*FtsZ assembly, to induce morphological changes and inhibit *Xoo* growth in vitro, and to prevent or ameliorate rice bacterial leaf blight in vivo. We also summarize and highlight key aspects of the structure–activity relationship (SAR) of the DMEP scaffold. A summary of the approach is presented in Figure 1, and the corresponding workflow of virtual screening and bioassay is outlined in Figure 2.

## 2. Results and Discussion

### 2.1. Design and Synthesis of Target Compounds

Encouraged by our previous work [23], DMEP was currently certified as a promising scaffold of *Xoo*FtsZ inhibitors, but discovering how to guide and prepare higher active compounds derived from DMEP quickly and with high efficiency is a crucial purpose of our current work. To maximize the identification of derivatives that would be effective, safe to non-target organisms, and easily synthesizable, we employed a ligand-based approach followed by reranking of molecular docking scores using structure-based virtual screening. Briefly, the structure of DMEP was submitted to DrugSpaceX and 100 drug-like DMEP analogs were downloaded and docked with reconstructed *Xoo*FtsZ using Sybyl-X 2.0 software. The top 10 analogs of DMEP were selected by ranking the docking scores, which were obtained for each analog in various positions, thereby providing an indication of the accuracy and stability of the docking simulations. Thus, the higher the score, the more stable was the predicted interaction. Notably, many of the selected compounds had similar characteristics, such as substitutions of the para and meta positions of the phenyl ring that could potentially increase the protein−compound interaction (Figure 3). Overall, these results predicted that 4′-substituted DMEP analogs would be easily synthesizable and may have better bactericidal properties than DMEP.

To evaluate the effects of the 4′-substituted DMEP analogues, a series of title compounds were synthesized, and their synthetic routes were displayed in Figure 1 and Figure 2. As shown in Figure 4, the substituent position of monoester derivatives was confirmed based on the chemical shifts of H-4 and OH-4′. Notably, for DMEP, the chemical shift of OH-4′ was identified at 5.41 ppm, and the chemical shift of H-4 was confirmed at 4.86–4.87 ppm. By contrast, the OH-4′ group of compounds **B_1_**, **B_2_** and **B_3_** were substituted by the acyloxy group or sulfuryl group, and the corresponding chemical shifts disappeared in the spectrum, respectively. Moreover, the chemical shift of H-4 of compounds **B_1_**, **B_2_**, and **B_3_** remained at 4.83 ppm. Thus, this obviously demonstrated that the compounds **B_1_**, **B_2_**, and **B_3_** were substituted by the acyloxy group or sulfuryl group at the OH-4′ group.

### 2.2. The Anti-Xoo Bioactivity of Title Compounds

The antibacterial potency of the title compounds was first evaluated by measuring the growth of *Xoo* in vitro in the presence of a range of compound concentrations (Table 1). Most of the compounds had low antibacterial activity and only compound **B_2_** exhibited moderate activity. Thus, the 50% effective concentration (EC_50_) for inhibition of *Xoo* growth was 153 mg L^−1^ for compound **B_2_** and >200 mg L^−1^ for the remaining DMEP derivatives, which compared with 39.7 mg L^−1^ for DMEP and 36.3 mg L^−1^ for the control antibacterial agent, bismerthiazol. To further examine the inhibitory activity of these compounds, we measured the GTPase activity of purified recombinant *Xoo*FtsZ in vitro in the presence of compound **B_2_** or the control GTPase inhibitor berberine (Table 2). Compound **B_2_** inhibited purified *Xoo*FtsZ GTPase activity by 54.8% at 200 µM and by 48.6% at 100 µM. Further screening yielded 50% inhibitory concentrations (IC_50_s) of 159.4 µM and 225.0 µM for compound **B_2_** and berberine, respectively. Thus, although compound **B_2_** inhibited *Xoo*FtsZ GTPase activity with slightly higher potency than berberine and DMEP (IC_50_ = 278.7 µM), as demonstrated in our previous study [23], compound **B_2_** was less potent than DMEP for inhibition of *Xoo* growth. One possible explanation for this apparent discrepancy may be the relatively poor aqueous solubility of compound **B_2_** compared with DMEP, which may have restricted the bactericidal activity of compound **B_2_** to a greater extent compared with its GTPase-inhibiting activity. However, the LogP values of the compounds, as determined with ChemDraw Professional 17.0, predicted that DMEP would have a lower cLogP value compared with compound **B_2_** (cLogP = 0.97 and 3.69, respectively). Taken together, these analyses indicated that compound B_2_ exerted moderate anti-*Xoo* activity and outstanding *Xoo* GTPase-inhibitory activity. Therefore, we selected compound **B_2_** for further analysis of its potential bactericidal activity and mechanism of action.

### 2.3. Investigation of Action Mechanism for Prepared Compound B_2_ Targeting XooFtsZ

We next examined the effects of compound **B_2_** on the morphology of *Xoo* cells using transmission electron microscopy (TEM) and fluorescence microscopy. Incubation of *Xoo* with 100 mg L^−1^ compound **B_2_** significantly increased the average *Xoo* cell length from 2.05 ± 0.27 µM at 0 h to 4.03 ± 2.20 µM and 5.86 ± 3.23 µM at 12 h and 24 h, respectively (Figure 5A). Similarly, fluorescence microscopy of *Xoo* cells labeled with the lipophilic dye FM 4-64 and the DNA-intercalating dye 4′-6-diamidino-2-phenylindole revealed the elongated and filamentous appearance of the cells after incubation with compound **B_2_** (Figure 5B), which confirmed the TEM results.

Direct binding between compound **B_2_** and recombinant *Xoo*FtsZ was evaluated by measuring the intrinsic fluorescence intensity of *Xoo*FtsZ before and after the addition of compound **B_2_**. As shown in Figure 5C, the emission fluorescence intensity decreased in the presence of compound **B_2_** in an increasing, concentration-dependent manner. The K_A_ of *Xoo*FtsZ–compound **B_2_** complex formation was calculated as 10^3.22^ M^−1^, which was similar to that of *Xoo*FtsZ–DMEP at 10^3.48^ M^−1^ (Table 3). Potential conformational changes in *Xoo*FtsZ triggered by compound **B_2_** binding were investigated using FT-IR. In the spectra shown in Figure 5D, 1600–1700 cm^−1^ represents the amide I band, which relates to the secondary structure of *Xoo*FtsZ. Compared with free *Xoo*FtsZ, complexes of *Xoo*FtsZ and compound **B_2_** exhibited peaks in the 1600 cm^−1^ to 1700 cm^−1^ region, suggesting that compound **B_2_** binding altered C-N stretching and N-H bending in *Xoo*FtsZ. The broader band at 3400 cm^−1^ also indicated that *Xoo*FtsZ–B_2_ complexes exhibited O-H and N-H stretching vibrations compared with free *Xoo*FtsZ. These interesting results suggested that compound **B_2_** binding to *Xoo*FtsZ changed the protein conformation, which may be responsible for the change in the biological activity of *Xoo*FtsZ.

Self-assembly of *Xoo*FtsZ was monitored by TEM and showed that, whereas free *Xoo*FtsZ formed single-stranded and uniform protofilaments, addition of compound **B_2_** to *Xoo*FtsZ resulted in fewer single-stranded protofilaments and an increase in disordered and disorganized protein aggregation compared with the control sample. This finding demonstrated that compound **B_2_** binding disorders the self-assembly of *Xoo*FtsZ via regulation of protein conformation, suggesting a mechanism for the inhibition of *Xoo*FtsZ GTPase activity.

Molecular docking is an increasingly common and effective approach for predicting possible binding modes of small molecules complexed with proteins [26,27,28]. Investigation of *Xoo*FtsZ–**B_2_** docking (Figure 6) showed that Asp38 and Arg205 were the main residues interacting with compound **B_2_** to form hydrogen bonds. Sulfur-X, alkyl, π–δ, and Van der Waals bonds interaction also appeared crucial for complex formation. π-Alkyl or alkyl interactions were observed between compound **B_2_** and Met32, Val33, Phe42, and Val40 residues; sulfur-X interaction was observed between compound **B_2_** and Met32; and π–δ bonding was observed between compound **B_2_** and Val40 (Figure 6). The docking scores are provided in Appendix A. Collectively, these molecular docking results showed that the docking score for *Xoo*FtsZ interaction was higher for compound **B_2_** than DMEP (6.34 vs. 5.92). These results further substantiated the results of our FT-IR spectra analysis.

### 2.4. Potential Mechanism of Action for 4′-Demethylepipodophyllotoxin (DMEP) Analogues

DMEP and its derivatives represent a sustainable natural bioresource with antifungal [29], anticancer [30,31], and antiviral [32] activities, among other biological properties. To begin the SAR of the DMEP scaffold and *Xoo*FtsZ activity, we tested several commercially available DMEP analogs and found that they all exhibited weak anti-*Xoo* activity in vitro compared with the parent compound (Table 4). Determination of the minimum inhibitory concentrations (MICs), which represent the lowest concentrations that inhibit *Xoo* growth, showed that DMEP and bismerthiazol both had MICs of 50 mg L^−1^, whereas the remaining analogs tested had much poorer anti-*Xoo* activities (MICs > 200 mg L^−1^). Despite this, examination of the effects of these compounds on *Xoo* cell morphology showed that several compounds, including teniposide and etoposide, induced cellular elongation similar to DMEP and compound **B_2_** (Figure 7). The binding parameters for these compounds and *Xoo*FtsZ were determined (Figure 8 and Table 5) and showed that the quenching mechanism between *Xoo*FtsZ and these compounds could format a weaker noncovalent complex than compound **B_2_**. The K_A_ values for the interactions between *Xoo*FtsZ and podophyllotoxin, picropodophyllotoxin, 4′-demethylpodophyllotoxin (DMEOP), deoxypodophyllotoxin, teniposide, and etoposide were 10^1.34^ M^−1^, 10^1.81^ M^−1^, 10^2.26^ M^−1^, 10^1.04^ M^−1^, 10^1.73^ M^−1^, and 10^2.50^ M^−1^, respectively, all of which were lower than the K_A_ of 10^3.48^ M^−1^ for DMEP–*Xoo*FtsZ. Overall, these results indicated that the hydroxyl group of DMEP was crucial for its anti-*Xoo* activity as well as for its interaction with *Xoo*FtsZ.

### 2.5. Outcome of SAR Study

To extend the SAR of compounds based on the DMEP, we systematically examined the antibacterial potency of DMEP analogs based on inhibition of *Xoo* growth in vitro. The results can be summarized as follows (Figure 9): (1) when the 4-position is in the S configuration, a bulky group at the 4-position was unfavorable to anti-*Xoo* activity: DMEP (EC_50_ = 38.7 mg L^−1^) > teniposide and etoposide (both EC_50_ > 200 mg L^−1^); (2) the S configuration of the hydroxyl group was excellent for anti-*Xoo* activity: DMEP (EC_50_ = 38.7 mg L^−1^) > deoxypodophyllotoxin and DMEOP (both EC_50_ > 200 mg L^−1^), which was in agreement with the docking results for these compounds (Figure 6); (3) the R configuration of the H group at the 2-position was beneficial for anti-*Xoo* activity: **A_1_** (EC_50_ > 200 mg L^−1^) < **B_2_** (EC_50_ = 153 mg L^−1^); and (4) the hydroxyl group at the 4′-position increased the anti-*Xoo* activity: DMEP (EC_50_ = 39.7 mg L^−1^) > **B_1_** (EC_50_ > 200 mg L^−1^) and **B_2_** (EC_50_ > 153 mg L^−1^).

### 2.6. In Vivo Trials against Rice Bacterial Leaf Blight Infected by Xoo

Encouraged by these in vitro results, we asked whether compound **B_2_**-mediated inhibition of *Xoo*FtsZ might provide an effective approach to controlling bacterial leaf blight diseases. Using pot experiments, we observed that compound **B_2_** had good curative activity against rice bacterial leaf blight and gave a control efficiency of 54.9% at 200 mg mL^−1^, which was better than both commercial TC (31.2%) and, as previously reported, DMEP (50.0%) [23]. Similarly, compound B_2_ had superior protective activity (48.4%) against bacterial leaf blight compared with either TC (30.4%) or DMEP (46.8%) [23]. Thus, targeting of bacterial FtsZ by compound **B_2_** holds promise for the management of plant bacterial diseases.

### 2.7. Assessment of Potential Risk of DMEP and Compound B_2_ through Phytotoxicity and Cytotoxicity Testing

Determining the potential off-target toxicity of novel agricultural and pest management agents is an important consideration in the development of safer and more environmentally responsible toxins. Therefore, we compared the potential phytotoxicity of DMEP and compound **B_2_** against rice plants, as previously described [33]. Notably, compound **B_2_** was non-toxic to rice plants at a concentration of 200 mg L^−1^, which was an effective dose for anti-*Xoo* activity in vivo. We also examined the cytotoxicity of DMEP and compound **B_2_** against two representative mammalian cell lines in vitro using a standard MTT cytotoxicity assay [34,35]. We tested the normal rat kidney cell line NRK-52E and the human non-small cell lung cancer cell line (A549), which was included because several DMEP analogs are already in clinical use as anticancer agents. Interestingly, compound **B_2_** was more cytotoxic than either DMEP or gefitinib, a small molecule clinical used for the treatment of lung cancer, against A549 cells, but was the least cytotoxic compound against NRK-52E cells (IC_50_ 60.8 µM compared with 30.8 µM and 21.0 µM for DMEP and gefitinib, respectively), and corresponding results was showed in Appendix A. Furthermore, to illustrate the druggability of compound **B_2_**, we submitted the structure of compound **B_2_** into the website http://www.swissadme.ch/index.php (accessed on 1 November 2021), and the corresponding results showed that compound **B_2_** met the drug-likeness rules, including Lipinski, Veber, Egan, and Muegge, with a bioavailability score of 0.55 [36]. Notably, these data showed that compound **B_2_** has high anti-*Xoo* activity, low phytotoxicity, high antiproliferative activity against the A549 cancer cell line, and low antiproliferative activity against the normal NRK-52E cell line.

## 3. Materials and Methods

### 3.1. Instruments and Chemicals

Instruments: NMR spectra of prepared title compounds were obtained on a Bruker Biospin AG-400 instrument (Bruker Optics, Ettlingen, Germany) using DMSO-*d_6_*/CDCl_3_ as solvent and tetramethylsilane as the internal standard; HRMS spectra were achieved using Waters Xevo G2-S QTOF MS (Waters MS Technologies, Manchester, UK). TEM images of *Xoo*’s morphological changes were visualized on a FEI Talos F200C electron microscope (FEI, Hillsboro, OR, USA) operating at a voltage of 200 kV. Fluorescence spectra data were performed on a FluoroMax^®^-4P (HORIBA Scientific, Paris, France). The FT-IR spectra data were recorded on a Nicolet iS50 instrument (Thermo Fisher Scientific, Waltham, MA, USA). Fluorescent images of *Xoo* cells were achieved using an Olympus-BX53-microscope (Olympus, Tokyo, Japan). The optical values were recorded on Cytation™5 multi-mode readers (BioTek Instruments, Inc., Winooski, VT, USA). Recombinant *Xoo*FtsZ was purified by a GE ÄKTA pure 25 system (GE Healthcare Bio-Sciences, Piscataway, NJ, USA).

Chemicals: All the chemicals were purchased from Bide Pharmatech Co., Ltd. (Shanghai, China) and Energy Chemical of Saen Chemical Technology Co., Ltd. (Shanghai, China). The Ni-NTA column (1 × 5 mL) and HiTrap desalting column (5 × 5 mL) were acquired from the GE Healthcare company (USA). IPTG (isopropyl *β*-D-thiogalactoside), HEPES, EDTA, disodium hydrogenphosphate, sodium dihydngen phoshate, imidazole, and NaCl were provided by the Bioengineering Co., Ltd. (Shanghai, China) and Solarbio Life Sciences & Technology Co., Ltd. (Beijing, China). GTP was ordered from ThermoFisher Scientific Vendor Co., Ltd. (Shanghai, China).

### 3.2. Experimental Section

The wild-type *Xanthomonas oryzae* pv. *oryzae* (*Xoo*) strain ZJ173 was kindly provided by Prof. Ming-Guo Zhou (Nanjing Agricultural University, Nanjing, China). The minimum inhibitory concentration (MIC) and in vivo of anti-*Xoo* bioactivity (in vitro and in vivo assay), and purification of recombinant *Xoo*FtsZ. The structures of the title compounds were characterized by ^1^H NMR, ^13^C NMR and HRMS, and corresponding data was provided as Appendix A. All of the above-mentioned experimental details can be found in Appendix A.

### 3.3. The Strategy of Structure-Based Virtual Screening (SBVS)

Initially, the amino acid sequence of *Xoo*FtsZ was achieved from the website of the national center for biotechnology information, and its three-dimensional structure was reconstructed through using multi-template modeling. Particularly, modeling *Xoo*FtsZ’s protein backbone dihedral angle parameters was further refined through the GROMOS 54A7 force field. These details can be found in our previous work [37].

In the second stage of the virtual screening, a structure-based virtual screening (SBVS) approach was carried out through using the database of DrugSpaceX [25]. Notably, more than 100 million chemical products bearing synthesizable and drug-like properties were provided in the DrugSpaceX database. Briefly, the structure of DMEP was submitted to the DrugSpaceX website (https://drugspacex.simm.ac.cn/, accessed on 1 October 2021), and one hundred DMEP analogues were visualized in the website. Thereafter, these DMEP analogues were downloaded as a subset and further used for the virtual screening. Subsequently, the automated protein preparation protocol was used for docking by operating Sybyl-X 2.0 software (Tripos Associates, Saint Louis, MO, USA). Finally, according to the results of the docking score, the top 10 compounds with the best scores were listed and ranked in Figure 3.

### 3.4. Determination of the Binding Constant (K_A_) of Compounds-XooFtsZ Interaction

The dissociation constants of compounds-*Xoo*FtsZ were determined by using typical fluorometric titration assays [38,39]. Briefly, 10 μM *Xoo*FtsZ was co-incubated with various concentrations (0, 2.5, 5.0, 7.5, 10.0, 12.5, 15.0, 17.5 and 20.0 μM) of test compounds in 20 mM phosphate buffer (pH 7.4) containing 150 mM KCl and 1 mM EDTA at 25 °C. Then, these samples were recorded using the FluoroMax^®^-4P instrument (Ex = 280 nm, slit widths = 3 nm). The corresponding binding constant (K_A_) of each sample was calculated by utilizing the Stern–Volmer method (F_0_/F = 1 + Kq τ0[Q] = 1 + Ksv [Q]) at 334 nm.

### 3.5. Morphological Studies Using Transmission Electron Microscopy (TEM)

*Xoo* cells (OD_595_ = 0.1) were co-incubated without/with 100 mg L^−1^ of compound **B_2_** in nutrient broth for 24 h in a shaker (180 rpm, 28 ± 1 °C). After that, these samples were covered with Formvar-carbon-coated copper grids and then negatively stained using 1% phosphotungstic acid. Finally, prepared samples were photographed by operating a transmission electron microscope (TEM), and the corresponding *Xoo* length of each sample was measured using Image*J* software (NIH Image, Bethesda, MD, USA) [40,41].

### 3.6. Fourier Transform Infrared (FT-IR) Spectroscopy Analysis

The FT-IR spectra analysis was carried out by referring to previously reported methods [42,43]. Briefly, 30 µM of *Xoo*FtsZ was mixed without/with 10 µM compounds in 20 mM phosphate buffer (pH 7.4) containing 150 mM KCl and 1 mM EDTA at 25 °C for 10 min. Then, 2 µL of treated sample was covered on the new KBr disc. Finally, the spectra of each sample were scanned under a certain condition (Scanning area: 500–4000 cm^−1^, scans: 32, resolutions: 4 cm^−1^). Particularly, the background spectrum was pre-recorded. The FT-IR spectra of each sample were yielded using a Nicolet iS50 instrument (Thermo Fisher Scientific, Waltham, MA, USA) (*n* = 2 for every group).

### 3.7. Fluorescence Patterns for the Xoo Cells Triggered by Compounds

*Xoo* cells were precultured in the above condition (2.5) and also displayed their morphological changes through using a BX53 fluorescence microscope. Briefly, the *Xoo* cells were fixed with 7% formaldehyde for 10 min and further washed with phosphate-buffered saline buffer (PBS, 10 mM, pH 7.3). Thereafter, these samples were stained with FM™ 4-64 dye solution (3 mg L^−1^) for 20 min and subsequently washed by phosphate-buffered saline buffer (PBS, 10 mM, pH 7.3). Finally, these samples were spread on a glass slide and then stained with DAPI solution (2 mg L^−1^) for fluorescence imaging [44,45].

### 3.8. Statistical Analysis

Statistical analyses were executed with one-way ANOVA by using SPSS 20.0 software. The Duncan (D) adjustment was performed to determine the significant difference between different treatments. Asterisks represented significant differences in comparison to control: (*) *p* < 0.05 and (**) *p* < 0.01. In the section of anti-*Xoo* bioassay in vivo, different uppercase letters following the control efficiency values illustrated that there was a significant difference (*p* < 0.05) among different treatment groups. The results were presented as means ± SD. 

## 4. Conclusions

As the cause of rice bacterial leaf blight, the vascular phytopathogenic bacterium *Xoo* is a major cause of reduced crop quality and quantity. Based on our previous work identifying DMEP as a novel chemical scaffold for *Xoo*FtsZ inhibitors, we used a combination of in silico, in vitro, and in vivo approaches to design and systematically test DMEP derivatives with potential anti-*Xoo* activity. Compound **B_2_** was validated as a potential *Xoo*FtsZ inhibitor with an IC_50_ (159.4 µM) lower than that of the parent DMEP (278.0 µM). We also showed that compound **B_2_** likely binds to *Xoo*FtsZ by interacting with residues Asp38, Arg205, Met32, Val33, Phe42, and Val40, and that the interaction disrupts FtsZ linear assembly and induces elongation of *Xoo* cells. Finally, we showed that compound **B_2_** displayed good curative and protective activities against rice bacterial leaf blight in pot studies but displayed low general phytotoxicity against rice plants and low cytotoxicity against mammalian cell lines. Taken together, our results identify compound **B_2_** as a promising FtsZ-targeting DMEP derivative that could be developed for the management of plant bacterial diseases.

## Data Availability

Not applicable.

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
