# Peer review of "Discovery of Epipodophyllotoxin-Derived B2 as Promising XooFtsZ Inhibitor for Controlling Bacterial Cell Division: Structure-Based Virtual Screening, Synthesis, and SAR Study"

_ijms, 2022, doi:10.3390/ijms23169119_

Round 1

Reviewer 1 Report

Figure 2. mistake in analogues word "ananlogues"

Author Response

Figure 2. mistake in analogues word "ananlogues"

Answer: We are grateful for your meticulous advice, and this mistake has been carefully corrected.

Reviewer 2 Report

It would be interesting to the readers to include the results of SWISSADME studies in the manuscript for the most potent compounds. The analysis can be in brief. The authors can look at the previously published papers involving SWISSADME studies.

Author Response

Answer: Thanks for the kind suggestions. The corresponding results were supplemented in revised menuscript. “Furthermore, to illustrate the druggability of compound B2, we submitted the structure of compound B2 into the website http://www.swissadme.ch/index.php, the correspongding results showed that compound B2 meeted the druglikeness rules including Lipinski, Veber, Egan, and Muegge, with bioavailability score of 0.55.”

Reviewer 3 Report

This referee focused on the SAR (docking) procedures. The results seem to be sound, but I do not find table S1 (should it be in the supporting information data?). The authors claim that the docking calculations are stable and fine. Many recent studies warn about the reproducibility of docking results across distinct platforms. I wonder if is possible to perform (repeat) some of the calculations using another platform. Will be the results (orderings, binding intensities,...) the same or having the same order across ligands? This should be interesting to know.

Author Response

Answer: Thanks for the kind suggestions. Firstly, we apologized this mistake, the Table S1 was added in revised supporting information. Secondly, as your wise judgement, we aim to provide a comprehensive SAR study of DMEP. Therefore, we preferentially choosed the  Sybyl-X 2.0 software, which was highly recognized for many years. Furthermore, we needed an effective comparation according to our previous work (Zhou, X.; et al, Ind. Crop. Prod. 2021, 174, 114182. https://doi.org/10.1016/j.indcrop.2021.114182. ). In addition, refering your meticulous advice and our experimence,  the results were analogical but not exactly same. In our future plan, we will compare docking scores through using various plantforms. Thanks.

Zhou, X.; Ye, H.J.; Gao, X.H.; Feng, Y.M.; Shao, W.B.; Qi, P.Y.; Wu, Z.B.; Liu, L.W.; Wang, P.Y.; Yang, S. The discovery of natural 4'-demethylepipodophyllotoxin from renewable dysosma versipellis species as a novel bacterial cell division inhibitor for controlling intractable diseases in rice. Ind. Crop. Prod. 2021, 174, 114182. https://doi.org/10.1016/j.indcrop.2021.114182.

Reviewer 4 Report

The manuscript by Song et al. reports the development of novel antibacterial agents based on previously discovered DMEP scaffold. Among several synthesized and commercially available DMEP analogs, compound B2 exhibited the improved IC50 values. The authors studied the potential mechanism of action by observing bacterial cell elongation using TEM. Moreover, they found that differences in association constants obtained by fluorescence spectroscopy match the trend in the observed biological activity. They identified several key interactions responsible for stabilization of protein-ligand complex using molecular docking. In vivo studies on rice bacterial leaf blight confirmed the improved activity of B2 compared with DMEP. Compound B2 was also non-toxic to rice plants and healthy human cell lines, and exhibited a notable anticancer activity against A549 cells. Authors also reported preliminary SAR study od DMEP analogs as XooFtsZ inhibitors.

The manuscript is well written and was easy to follow. In my opinion, these results are significant and should be published in IJMS.

Some minor changes and suggestions are listed below.

Figure 2 – ananlogues – change to analogs; DrugSpaceX databases – change to DrugSpaceX database

Line 95: More negative value of docking score indicates stronger interaction, not higher values. All the scores in Figure 3 probably have – prefix, so the best one is –11. Also the unit is missing and it is probably kcal/mol.

Figure 3 caption – “Top 10 compounds resulting from docking scores in silico predictions” should be simplified by removing either docking scores or in silico predictions.

Figure 5 D – instead of entire wavenumber range, authors may show only region relevant to amide I band.

Line 194: What are V-H bonds? Did authors mean Van der Waals bonds?

Line 217: “could format a weaker noncovalent static complex” should be rewritten. Authors may omit static from this sentence.

Line 337: softerware – correct the spelling

Author Response

Figure 2 – ananlogues – change to analogs; DrugSpaceX databases – change to DrugSpaceX database

Answer: We are grateful for your meticulous advice, and the corresponding mistakes have been carefully corrected.

Line 95: More negative value of docking score indicates stronger interaction, not higher values. All the scores in Figure 3 probably have – prefix, so the best one is –11. Also the unit is missing and it is probably kcal/mol.

Answer: We are grateful for your meticulous advice. For the Figure 3, we choose the Sybyl-X 2.0 software to give the docking scores. Total_Score = The total Surflex-Dock score expresses as -log(Kd). Namely, Surflex-Dock scores are expressed in -log10(Kd) units to represent binding affinities. Surflex-Dock uses an empirically derived scoring function that is based on the binding affinities of protein-ligand complexes and on their X-ray structures. Therefore, the best one has highest score.

The Surflex-Dock scoring function and the protein-ligand complexes used to calibrate it are described in Jain, A.N. "Scoring noncovalent protein-ligand interactions: A continuous differentiable function tuned to compute binding affinities." J. Comput. Aided-Mol. Des. 1996, 10, 427-40.

Figure 3 caption – “Top 10 compounds resulting from docking scores in silico predictions” should be simplified by removing either docking scores or in silico predictions.

Answer: Thanks for the kind suggestions. The Figure 3 caption was revised as “Top 10 compounds resulting from docking scores”.

Figure 5 D – instead of entire wavenumber range, authors may show only region relevant to amide I band.

Answer: We are grateful for your meticulous advice. Generally, an entire wavenumber range of IR spectra was contributed to comprehensive analysis. In this work, the results of Figure 5 D showed that a distinct amide I band could be found. But not all results are same. Therefore, in our future plan, we will show more details about these data.

Line 194: What are V-H bonds? Did authors mean Van der Waals bonds?

Answer: We apologized this mistake. The “V-H bonds” was revised as “Van der Waals interaction”.

Line 217: “could format a weaker noncovalent static complex” should be rewritten. Authors may omit static from this sentence.

Answer: Thanks for the kind suggestions. The “static” word was removed in revised manuscript.

Line 337: softerware – correct the spelling

Answer: We apologized this mistake. The “softerware” was revised as “software” in revised manuscript.